# Common Variants in One-Carbon Metabolism Genes (*MTHFR*, *MTR*, *MTHFD1*) and Depression in Gynecologic Cancers

**DOI:** 10.3390/ijms241612574

**Published:** 2023-08-08

**Authors:** Piotr Pawlik, Grażyna Kurzawińska, Marcin Ożarowski, Hubert Wolski, Krzysztof Piątek, Radosław Słopień, Stefan Sajdak, Piotr Olbromski, Agnieszka Seremak-Mrozikiewicz

**Affiliations:** 1Division of Gynecological Surgery, Poznan University of Medical Sciences, Fredry 10, 61-701 Poznan, Poland; pitpawlik@gmail.com (P.P.); ssajdak@ump.edu.pl (S.S.); olbromski.piotr@gmail.com (P.O.); 2Division of Perinatology and Women’s Diseases, Poznan University of Medical Sciences, Fredry 10, 61-701 Poznan, Poland; gkurzawinska@ump.edu.pl (G.K.); asm@data.pl (A.S.-M.); 3Laboratory of Molecular Biology, Division of Perinatology and Women’s Diseases, Poznan University of Medical Sciences, Fredry 10, 61-701 Poznan, Poland; 4Department of Biotechnology, Institute of Natural Fibres and Medicinal Plants—National Research Institute, Wojska Polskiego 71B, 60-630 Poznan, Poland; 5Institute of Medical Sciences, Academy of Applied Sciences, Kokoszków 71, 34-400 Nowy Targ, Poland; hubertwolski@wp.pl; 6Department of Gynecology and Obstetrics, University of Zielona Gora, Licealna 9, 65-417 Zielona Gora, Poland; krzysztofpiatek.jr@icloud.com; 7MedicaNow Gynecological Endocrinology and Menopause Clinic, Piątkowska 118, 60-648 Poznan, Poland; asrs@wp.pl

**Keywords:** polymorphism, *MTHFR*, MTR, *MTHFD*, gynecologic cancers, depression

## Abstract

We investigated the association between methylenetetrahydrofolate reductase (gene *MTHFR* 677C>T, rs1801133), 5-methyltetrahydrofolate-homocysteine methyltransferase (*MTR* 2756A>G, rs1805087), and methylenetetrahydrofolate dehydrogenase, cyclohydrolase and formyltetrahydrofolate synthetase 1 (gene *MTHFD1* 1958G>A, rs2236225)—well-studied functional variants involved in one-carbon metabolism—and gynecologic cancer risk, and the interaction between these polymorphisms and depression. A total of 200 gynecologic cancer cases and 240 healthy controls were recruited to participate in this study. Three single-nucleotide variants (SNVs) (rs1801133, rs1805087, rs2236225) were genotyped using the PCR-restriction fragment length polymorphism method. Depression was assessed in all patients using the Hamilton Depression Scale. Depression was statistically significantly more frequent in women with gynecologic cancers (69.5% vs. 34.2% in controls, *p* < 0.001). *MTHFD1* rs2236225 was associated with an increased risk of gynecologic cancers (in dominant OR = 1.53, *p* = 0.033, and in log-additive models OR = 1.37, *p* = 0.024). Moreover, an association was found between depression risk and *MTHFR* rs1801133 genotypes in the controls but not in women with gynecologic cancers (in codominant model CC vs. TT: OR = 3.39, 95%: 1.49–7.74, *p* = 0.011). Cancers of the female reproductive system are associated with the occurrence of depression, and ovarian cancer may be associated with the rs2236225 variant of the *MTHFD1* gene. In addition, in healthy aging women in the Polish population, the rs1801133 variant of the *MTHFR* gene is associated with depression.

## 1. Introduction

One-carbon metabolism (OCM) involves three interconnected pathways: the folate cycle; methionine remethylation; and trans-sulfuration pathways. Methionine (Met) and folic acid (FA, B9) are the key components of OCM, providing the methyl groups for numerous methyl transferase reactions via the major cellular methyl donor, S-adenosylmethionine (SAM). A derivative of vitamin B9 is a tetrahydrofolate (THF), which participates in OCM and in the synthesis of several amino acids such as serine and methionine, purines, and pyrimidine (thymine),. The single-carbon transfer reactions are important in the metabolism and regulation of gene expression [1,2]. The folate cycle is composed of cytoplasmic and mitochondrial compartments. In the cytoplasmic compartment, THF is the substrate for MTHFD1—an NADP-dependent protein possessing three distinct enzymatic activities: methylenetetrahydrofolate dehydrogenase (EC 1.5.1.5); methenyltetrahydrofolate cyclohydrolase (EC 3.5.4.9); and formate–tetrahydrofolate ligase (EC 6.3.4.3). Each of these activities catalyzes one of the three following reactions in which tetrahydrofolate derivatives are formed: 10-formyl-THF, necessary for purine synthesis; 5,10-methenyl-THF, for the synthesis of deoxythymidine monophosphate (pyrimidine); 5,10-methylene-THF, for de novo dTMP synthesis. Methylenetetrahydrofolate reductase (MTHFR; EC 1.5.1.20) converts 5,10-methylene-THF to 5-methyl-THF, which is catalyzed by methionine synthase (MTR; EC 2.1.1.13), using vitamin B12 as a cofactor for the remethylation of homocysteine (Hcy) to methionine [3,4,5]. Hcy is metabolized by three reactions: the trans-sulfuration pathway; and two methylation reactions. The first methylation reaction requires the activity of methionine synthase and methionine synthase reductase, and the second is catalyzed by betaine homocysteine S-methyltransferase (BHMT; EC 2.1.1.5). The trans-sulfuration pathway begins with the vitamin B6-dependent enzyme beta cystathionine synthase (CBS; EC 4.2.1.22), which permanently removes homocysteine from the methionine cycle and initiates the synthesis of cysteine and glutathione (Figure 1).

Levels of the plasma intermediate sulfur-containing amino acid Hcy are strongly influenced by diet, as well as by genetic factors. Much effort has been devoted to studying the effects of single nucleotide variants (SNVs) in enzymes of the OCM pathway. One of the most studied variants in the *MTHFD1* gene is 1958G>A (R653Q, rs2236225), located within the 10-formate-tetrahydrofolate ligase domain, which may modulate biosynthesis of thymidylate, purine nucleotides, and methionine, affecting DNA methylation [6].

The polymorphism 677C>T (rs1801133) of the *MTHFR* gene results in decreased enzyme activity, which leads to an increase in 5,10-MTHF and a decrease in 5-MTHF. This frequently studied variant is associated with the risk of various gynecological neoplasms [7]. The variant 2756A>G (rs1805087) of the MTR gene is less studied, but it is also associated with the risk of various cancers (e.g., colorectal or thyroid) [8,9].

The hallmark of cancer cells is an altered metabolism, and one-carbon metabolism is associated with therapies used in the treatment of various malignant neoplasms. Many studies have proven that there is a close link between hyperhomocysteinemia and cancer. Higher levels of plasma homocysteine have been observed in cancer patients, and polymorphisms in the enzymes involved in the Hcy detoxification pathways have close clinical ties to several cancer types [10]. There is also evidence that diet and nutrition are modifiable risk factors for several cancers [11]. It is essential to expand our knowledge on the mechanisms of regulation of one-carbon metabolism, as more and more studies are confirming its key role in cancer [12]. Dietary folates are essential in many key metabolic processes, including many amino acid reactions, e.g., breakdown of histidine, reduced levels of which have been reported in cervical, lung, and ovarian cancer [13,14,15,16]. All women are at risk for gynecologic cancers, and risk increases with age. Declines in estradiol across stages of the menopause transition may lead to elevations in Hcy and cysteine, both associated with oxidative damage and metabolic disorders, which may lead to carcinogenesis [17]. Studies have reported on the probable association of endogenous sex hormones and homocysteine levels [18,19]. It has been shown that men have higher total homocysteine (tHcy) concentrations than women, and this difference is attenuated after menopause. Moreover, tHcy levels tend to be lower during pregnancy, which is associated with high endogenous estrogen levels [20,21].

Folate deficiency is also common in patients with depressive disorders [22]. SAM dependent methylation is crucial for the generation of catecholamine neurotransmitters, phospholipids, and myelin. The metabolism of tetrahydrobiopterin, a key co-factor for the synthesis of serotonin and catecholamine neurotransmitters, is regulated by tetrahydrofolate (THF), a derivative of the folate pathway. Malnutrition and related folic acid deficiencies can also be the result of poor diet in both cancer patients and depressive disorders. Therefore, disturbances in the OCM pathway can be primary or secondary in both diseases.

A systematic review of the literature on the incidence of depression and anxiety in patients with ovarian cancer (sample of 3623 patients) by stage of treatment suggests that the prevalence of depression and anxiety in these women, across the treatment spectrum, is significantly higher than in the population of healthy women [23]. Effective screening and treatment of depression symptoms may have important oncological consequences. In a meta-analysis, Walker et al. [24] examined whether depression is associated with worse survival in people with cancer. They analyzed data on 20,582 patients with breast, colorectal, gynecological, lung, and prostate cancers from Scotland, United Kingdom. Major depression was associated with worse survival for all cancer types. For gynecological cancers hazard ratio was 1.36 (95% CI = 1.08–1.71) [24].

Therefore, the aim of the present study was to investigate the effects of well-studied functional variants causing non-synonymous amino acid changes in *MTHFR*, *MTR*, and *MTHFD1* genes from the one-carbon metabolism pathways on the occurrence of gynecologic cancers and major depressive disorder in Polish women from Wielkopolska.

## 2. Results

### 2.1. Characteristics of Study Patients

A total of 200 unrelated women with gynecologic cancers ranging in age from 18 to 90 years (mean ± SD; 60.40 ± 12.17) and 240 controls ranging from 43 to 75 years old (mean ± SD; 60.10 ± 7.82) were enrolled in the hospital-based case-control study. In the gynecologic cancers group, 128 women had ovarian cancer, 48 had endometrial cancer, and 24 had cervical cancer. Figure 2 shows the percentage distribution of the study population.

Mean age was comparable between cancer cases and control women (*p* = 0.192). In the group of women with cancer, the most common was ovarian cancer (64%, mean age ± SD; 59.6 ± 11.8 years), followed by endometrial (24%; 61.8 ± 13.6 years) and cervical cancer (12%; 55.3 ± 11.9 years). Most patients had cancer for less than one year (72.30%), 19.25% had it for longer, and 8.45% had more than 5 years of cancer duration. On average, cancer was diagnosed in a patient aged 59 years old. Menopause occurred in 68.3% of healthy controls and 99.0% of women from the study group (in 50 women from this group, it was induced). Depression (defined as above 7 points on the Hamilton Depression Rating Scale) was statistically significantly more frequent in women with cancer (69.5% vs. 34.2% in controls, *p* < 0.001). The median (IQR) of the Hamilton Depression Rating Scale in women with depression was statistically significantly higher in the group with cancer and was 13 [10,11,12,13,14,15,16,17,18,19] vs. 12 [10,11,12,13,14] in the control group (*p* = 0.008) (Table 1).

A detailed assessment of depression in patients based on the Hamilton Depression Rating Scale scores for the groups of gynecological cancers and in the controls is presented in Table 2.

### 2.2. Association between Studied Polymorphisms and Malignant Neoplasms of Female Genital Organs

The SNPs rs1801133, rs1805087, and rs2236225 in the *MTHFR*, *MTR*, and *MTHFD1* genes were successfully genotyped in 200 patients and 240 healthy control subjects. The genotype distribution of these three polymorphisms in cases and controls was consistent with the Hardy–Weinberg equilibrium. Minor allele frequencies for cases and controls are shown in Table 3. Only for *MTHFD1* in the rs2236225 locus did we observe higher frequency of the A allele in women with cancers—0.420% vs. 0.346% in controls (OR: 1.369; 95% CI: 1.04–1.80; *p* = 0.023, *p*_corr_ = 0.071).

The genotype frequency distribution of rs1801133, rs1805087, and rs2236225 and their association with gynecologic cancers risk are shown in Table 4.

The *MTHFD1* rs2236225 AA genotype was associated with an increased risk of cancers (GG vs. AA, OR = 1.83, 95%: 1.02–3.28, *p* = 0.073; in dominant model GG vs. GA-AA, OR 1.53, 95%: 1.03–2.25, *p* = 0.033, AIC = 605.8; and in log-additive model OR = 1.37, 95%: 1.04–1.81, *p* = 0.024, AIC = 605.2). Allele A *MTHFD1* rs2236225 was more common in cancer patients than in controls (42.0% vs. 34.6%, OR = 0.730, 95% CI: 0.55–0.96, *p* = 0.024) (Table 3, Figure 3). No statistically significant associations were observed between *MTHFR* rs180133 and *MTR* rs1805087 variants and the risk of cancers.

When analyzing the frequency of genotypes in each tumor group, no statistically significant differences were found between the studied groups (Table 5).

When comparing cancer types with controls, statistically significant differences were observed only between ovarian cancer and the control group (GG vs. GA: OR = 1.71, 95% CI: 1.06–2.76, GG vs. AA: OR = 2.06, 95% CI: 1.05–4.01, *p* = 0.036, AIC = 474.9; for dominant model: OR = 1.78, 95% CI: 1.13–2.81, *p* = 0.012, AIC = 473.2; and log-additive: OR = 1.49 95% CI: 1.08–2.04, *p* = 0.014, AIC = 473.5).

### 2.3. Association between Studied Polymorphisms and Depression

The comparison of the frequency of genotypes between women with depression (*n* = 221) and women without depression (*n* = 219) indicated only the possible association of the rs1801133 variant of the *MTHFR* gene with the occurrence of depression (*p* = 0.076 in codominant model). We conducted stratification analysis by depression status in the case and control groups (Table 6), and this revealed an association between depression risk and MTHFR rs1801133 genotypes in the controls (in codominant model CC vs. TT: OR = 3.39, 95%: 1.49–7.74, *p* = 0.011) but not in women with gynecologic cancers. In the group of healthy women, the T allele was observed in 42.7% women with depression and in 28.8% without depression (*p* = 0.002, OR = 1.84, 95% CI: 1.24–2.73) (Figure 4). We did not observe statistically significant differences after stratification by cancer types.

### 2.4. SNP–SNP Interaction

We used multifactor dimensionality reduction (MDR 3.0.2) software for gene–gene interactions. The results of the interactions are presented in Table 7. The best single-locus model to predict gynecologic cancers was rs2236225 (testing accuracy, 0.5496; *p* = 0.033; cross-validation consistency, 10/10). The best two-locus model was a combination of rs1801133 and rs2236225, with the testing accuracy of 0.5208 and cross-validation consistency of 8/10.

In Figure 5, the graph shows the interactions between these SNPs. The largest main effect with higher information gain (IG) was observed for *MTHFD1* rs2236225 (0.86%), with 53.86% accuracy for this model. Analysis of the dataset of gynecologic cancers and controls revealed synergistic interactions between *MTHFD1* rs2236225 and *MTHFR* rs1801133 (IG = 0.06%) and MTHFR rs1801133 and MTR rs1805087 (IG = 0.08%). However, between *MTHFD1* rs2236225 and *MTR* rs1805087, IG was −0.04, which revealed negative entropy, indicating independence or redundancy.

## 3. Discussion

Even though major depressive disorder is common among cancer patients and its occurrence is higher than in the general population, it is often neglected. It is normal that due to changes in life plans, finances, and fear of pain and death, cancer patients are more likely to experience sadness and grief. However, depression is associated with a worsening of their condition and often with non-adherence to treatment [25,26]. High levels of homocysteine have been linked to both depression and cancers, but studies are not always consistent. Plasma homocysteine concentrations are determined via genetic variants in the enzymes involved in homocysteine metabolism or by nutritional deficiencies in vitamin cofactors. Moreover, Hcy level increases with age, at least in part due to lowered nutritional absorption and decreased metabolic function with advanced age [27,28,29]. Serum homocysteine concentrations were also progressively higher across menopausal stages [30]. Pathogenic levels of circulating Hcy, known as hyperhomocysteinemia (HHcy), are implicated in the induction of inflammatory determinants including the expression of adhesion molecules, leukocyte adhesion, endothelial dysfunction, oxidative stress, and reduced nitric oxide bioavailability. Despite the fact that many studies have been carried out, unfortunately, the molecular mechanisms underlying the adverse effect of HHcy have not been fully elucidated [31,32,33]. Emerging evidence suggests that not only hyperhomocysteinemia but also low levels of cysteine are associated with various diseases. Within the body, cysteine catabolic pathways are involved in the synthesis of coenzyme A, glutathione, taurine, and oxidized and reduced inorganic sulfur. It is known that cysteine plays an important role in decreasing the risk of cancer, diabetes, respiratory system-related problems, and influenza, but it also helps in improving fertility by reducing the reactive oxygen species level and increasing sperm motility and count [34,35]. N-acetylcysteine (NAC) is a precursor of cysteine; given as a supplement, it appears to be promising in the treatment of several psychiatric disorders [36,37]. Recently, Cheng et al. [38] analyzed, in a large study, the association of OCM genetic variants with blood biomarkers in postmenopausal women. They observed that significantly lower plasma cysteine concentrations were associated with MTHFD1 polymorphisms rs2236224 (G>A) and rs2236225 (R653Q, G>A) in non-Hispanic white women (−1.0%; 95% CI: −1.9%, −0.2%, per variant allele A; FDR-adjusted *p* = 0.078) [38].

The role of OCM gene polymorphisms in cancer patients has often been the subject of research. The MTHFR 677C>T polymorphism has been the most studied, but the conclusions are still controversial. In a meta-analysis, Karimi-Zarchi et al. [39] evaluated the association between MTHFR 677C>T variant and risk of ovarian and cervical cancers. A total of 27 case-control studies, including 4990 cases and 7730 controls, were selected. The analysis showed that 677C>T polymorphism of the *MTHFR* gene may not play a role in development of ovarian and cervical cancers in the general population. In our study, we also did not detect an association of this variant with any of the examined gynecological tumors. Nevertheless, in an earlier meta-analysis (2017), He and Shen found a possible association of this SNP with ovarian cancer in Asians, but not in Caucasians. The same study showed a strong relationship between the MTHFR 677C>T polymorphism in Caucasian people and breast cancer [40]. It should be noted that the frequency of the MTHFR 677T allele varies across ethnic groups and regions, which may result in differences in the studies. According to Phase 3 of the 1000 Genomes Project, the global population has a frequency of about 25%. The population with the highest frequency of the T allele is Hispanics (47%), followed by Europeans (36%), East Asians (30%), South Asians (12%), and Africans (9%) [41].

Interestingly, there are very limited data on the association of *MTR* and *MTHFD1* genetic variants with gynecological malignancies. We found an association of the MTHFD1 rs2236225 (R653Q) variant with ovarian cancer. However, in our study undertaken ten years ago, we did not observe an association of this SNP with the incidence of ovarian cancer and uterine cervical carcinoma, despite studies conducted in the same population of women [42,43]. Moreover, inhomogeneous results were obtained by Kelemen et al. [44], investigating the relationship between the rs1950902 (401G>A) variant of the *MTHFD1* gene and ovarian cancer. Initial observations showed associations between five SNPs of the OCM genes and the risk of ovarian cancer. However, in a re-examination, they were unable to confirm this association. The authors indicate the possibility of the influence of other SNPs in strong linkage disequilibrium with the studied gene variants on the relationship with ovarian cancer [44,45]. A relationship between the 401G>A, rs1950902 polymorphism of the *MTHFD1* gene, and ovarian cancer was also not found in a meta-analysis of 16 independent studies including 5195 cases and 9276 controls [46]. Moreover, through a meta-analysis, the impact of two SNPs in *MTHFD1* (1958G<A and 401G>A) on the risk of various cancers was investigated. For G1958A, a reduced risk of cancer was found in patients with acute lymphoblastic leukemia in Asians, and for G401A, the data showed an association with a reduced risk of colon cancer [6]. A genetic predisposition to ovarian cancer is suspected. Risk of becoming sick increased in women with a family history of breast, uterine, or ovarian cancer in a mother or sister [47]. Many studies focus on explaining the relationship between polymorphisms related to folate metabolism and the increased risk of ovarian cancer, but the presented results remain controversial [39,48]. A cytoplasmic trifunctional enzyme MTHFD1 catalyzes reactions in the pathway of conversion of tetrahydrofolate (THF), the active form of folate, into substrates essential for the de novo purine and thymidylate synthesis. Cancer cells rely on nucleotide metabolism, and the folate cycle is a necessary pathway for tumor growth. Moreover, the MTHFD1 indirectly provides one-carbon units for methylation reactions by the synthesis of 5,10 methylene-THF. The 1958G>A MTHFD1 variant may influence DNA synthesis reactions and cell development, eventually affecting carcinogenesis. This variant is in the MTHFD1 synthetase domain, which produces 10-formyltetrahydrofolate (formyl-THF) for the synthesis of purines from formate and THF. The 1958G>A polymorphism has been studied as associated with cancer, although with not univocal results. Wang and colleagues showed that the polymorphic 1958AA variant was linked to a significantly increased risk for gastric cancer as compared with the 1958GG or 1958AG genotypes [49], whereas Moruzzi showed that the 1958AA genotype was significantly less frequent among cancer patients and related to 75% reduction of colon cancer risk [50]. The 1958AA increases the thermostability of the protein, which changes its metabolic activity and impairs the synthesis of purines, which may hinder tumor growth and progression and have a protective effect in cancer. Moreover, the increased production of formyl-THF by the MTHFD1 synthetase may increase the expression of the mitochondrial enzyme MTHFD2 [50,51]. MTHFD2 is expressed in the developing embryo and is absent in most adult tissues [52,53]. Interestingly, markedly elevated expression of MTHFD2 was identified in many cancers and correlates with poor survival in breast cancer patients [54,55]. Recently the role of MTHFD2 in ovarian cancer was investigated by Cui et al. [56]. This study found that MTHFD2 is highly expressed in ovarian cancer, as well as an indispensable risk factor for poor prognosis in patients with ovarian cancer. Moreover, it has a role in malignancy mainly through the MOB1A signaling, which is a potential target for treating ovarian cancer [56].

Still little attention is paid to research on the impact of serious illnesses on patients’ mental health. Studies in animal models and human cancer cells grown in culture suggest that durable chronic stress accelerated tumorigenesis and progression, which is unfavorable for clinical outcomes of cancer patients [57,58]. Stress plays a fundamental role in many cases of depression, which is a common problem all over the world. Depression includes persistent sadness or loss of interest and pleasure accompanied by symptoms such as disturbed sleep or appetite, guilt or low self-esteem, fatigue, poor concentration, and difficulty making decisions. The illness can be long-term and affect both the mind and the body. More women are affected by depression than men, and its etiology is multifactorial, resulting from many interactions of social, psychological, and biological factors [59,60]. Research showed that abnormal folate metabolism associated with the presence of mutated alleles of the 677C>T MTHFR polymorphism is associated with increased risk for depression [61,62,63]. Folate is essential for the biosynthesis of neurotransmitters associated with mood, stress, motivation, and cognitive performance, such as serotonin (5-HT), noradrenaline (NA), and dopamine (DA) [64]. Moreover, folate deficiency affects that homocysteine levels are higher, and research has suggested a significant positive association between homocysteine levels and depressive symptom severity [65]. Receiving a cancer diagnosis is a difficult situation for anyone, and the disease affects all aspects of a patient’s life. The incidence of mental disorders in patients with cancer is very high (30–60%) [66,67]. However, some patients do better psychologically than others. In our study, depression (as measured by the Hamilton Depression Scale) was statistically significantly more common in women with cancer, but one third of them were not diagnosed with depression. This may be related to resilience, i.e., the ability of an individual to cope with suffering and adapt to difficult events [68,69]. A multi-center study conducted in ten cancer centers in Germany tried to determine how resilience is related to different demographics, other psychological factors, and different aspects of lifestyle (diet and physical activity). Based on the results of the Adolescents’ Food Habits Checklist (AFHC), they found a statistically significant positive relationship between resilience and eating habits in cancer patients (r = 0.117, *p* = 0.018) [70].

Malnutrition is present with the diagnosis of cancer in about 15–40% of cases, and this incidence increases during treatment, characterizing 40–80% of the patients in this phase. About 10–20% of cancer deaths are thought to be related to tumor- or treatment-induced malnutrition [71,72,73]. Sánchez-Torralvo et al. [74] observed a relationship between malnutrition and the presence of symptoms of anxiety and depression in hospitalized cancer patients. After controlling for potential confounders (age, sex, and cancer stage), malnourished patients were 1.98 times more likely to present anxious symptomatology (95% CI 1.01–3.98; *p*  =  0.049) and 6.29 times more likely to present depressive symptomatology (95% CI 1.73–20.47; *p*  =  0.005) [74].

Mutations in genes often occur during cancer progression, but the impact of a single nucleotide polymorphism in the genome on cancer predisposition is small. Inheritance of multifactorial diseases does not occur, but it is known that relatives may be at higher risk of developing the disease. Statistical analysis of interactions between different loci is expected to clarify the etiology of multifactorial diseases. The effect of one locus may be ineffective alone or masked by effects at another locus, but the combined result of several SNPs may be significant. Thus, interactions play an important role in the study of gene function and may improve understanding of their importance in biochemical pathways. Some papers also analyze SNP–SNP interactions in relation to the etiology of gynecological cancers. Interactions of CYP1 gene variants [75] and the X-ray repair of cross-complement 1 (XRCC1), tumor protein p53 (TP53), and fibroblast growth factor receptor 3 (FGFR3) gene polymorphisms [76] were significantly associated with the risk of cervical cancer in Chinese women. Moreover, in Chinese women, interactions of genes related to the metabolism of one-carbon units were analyzed. The joint impact of *MTRR* rs162036 and *MTR* rs1805087, *MTHFR* rs1801131 and *MTHFR* rs1801133, and folate and *MTHFR* rs1801133 may contribute to breast cancer risk [77].

Studies of the OCM genes in depression have produced many publications with conflicting results. Most studies concerned the *MTHFR* 677C>T gene variant. Several meta-analyses have found a relationship between the rs1801133 *MTHFR* variant and depression [62,78,79,80] but not all [81,82,83]. Moreover, studies have found associations between elevated homocysteine levels and low folate or vitamin B12 and depression [84,85,86]. However, a recent study of plasma homocysteine concentrations and depression in twins argued against a causal role of homocysteine in the development of depression [87]. Our study shows that neoplastic diseases are strongly associated with the occurrence of depression. We observed the influence of the *MTHFR* 677TT genotype on the incidence of depression, but this relationship was only found in the control group. It is likely that the main factors involved in the pathogenesis of depression in cancer patients are not genetic variants but psychological factors and pain. Even if there is an association between the 677C>T variant of the *MTHFR* gene and depression in the general Polish population, it has not been observed in cancer patients. Certainly, further research is needed in order to better unravel the role of the *MTHFD1* 1958 A>G variant in ovarian cancer risk, as this protein may be a potential therapeutic target for future ovarian cancer therapies.

## 4. Materials and Methods

### 4.1. Patient Selection

The case-control study was conducted at the University Hospital of Medical Sciences in Poznan, Poland, between 2018 and 2021. It was approved by the Local Bioethical Committee at Poznan University of Medical Sciences (No: 1128/18, from 7 November 2018) and conformed to the guidelines of the Declaration of Helsinki. Consecutive patients diagnosed with histologically recognized cancers of female reproductive organs according to the International Federation of Gynecology and Obstetrics (FIGO) were recruited to the study in the Clinic of Gynecological Surgery as cases. Healthy individuals, age matched and recruited in the same clinic, were included as a control group. Patient inclusion criteria were as follows: 18 or more years of age; no history of prior mental disorder or dementia; no abuse of alcohol or drugs; adequate knowledge of the Polish language and satisfactory level of communication; and consent to participate in the study. Overall, the study group consisted of 200 women with gynecologic cancers and 240 controls, all from the Greater Poland Voivodship. Baseline clinical characteristics were extracted from the medical records.

### 4.2. Anxiety Evaluation

Depression was assessed by the attending physician prior to blood sampling using the Hamilton Depression Rating Scale (HRSD). Criteria were adopted from the National Institute for Health and Clinical Excellence in the UK, which established depression levels in relation to 17 HRSD items compared to those suggested by the American Psychiatric Association (APA) [88]. The following categories were estimated: not depressed: 0–7; mild (subthreshold): 8–13; moderate (mild): 14–18; severe (moderate): 19–22; very severe (severe): >23.

### 4.3. Sample Collection for Genetic Testing and DNA Extraction

All patients and controls signed informed consent for genetic testing, in which the study management was described. Genomic DNA was extracted from whole blood samples stored in S-Monovette EDTA-coated tubes (Sarstedt, Nümbrecht, Germany) using the QIAamp DNA Mini Kit according to the manufacturer’s instructions (Qiagen GmbH, Hilden, Germany). DNA concentration and quality were determined spectrophotometrically using a NanoDrop 2000c (Thermo Fisher Scientific, Waltham, MA, USA). Isolated DNA was stored at −80 °C until analysis.

### 4.4. DNA Amplification and Genotyping

Basic information on selected polymorphic variants are presented in Table 8. Genotyping was performed in the Molecular Biology Laboratory of Poznan Medical Science University via polymerase chain reaction and restriction fragment length polymorphism (PCR-RFLP). The primers published by Frosst et al. [89], Hormon et al. [90], and Hol et al. [91] and the restriction enzymes HinfI (EURx, Gdansk, Poland), BsuRI (Thermo Fisher Scientific, Waltham, MA, USA), and MspI (EURx, Gdansk, Poland) for *MTHFR* (rs1801133), *MTR* (rs1805087) and *MTHFD1* (rs2236225), respectively, were used. Products were analyzed via electrophoresis on 2% agarose gel with Midori Green Advanced DNA Stain (Nippon Genetics, Düren, Germany). Positive and negative controls were included in each reaction as quality control, and for accuracy of genotyping, 90% of samples were repeated. The concordance between the original and the duplicate samples for all the analyzed SNVs was 100%.

### 4.5. Statistical Analysis

The normal distribution was tested using the Shapiro–Wilk test. Quantitative variables with Gaussian distribution were expressed as mean ± standard deviation (SD). Categorical variables are shown as number (percentages). The case and control populations were tested for the Hardy–Weinberg equilibrium (HWE). Multiple inheritance models (codominant, dominant, recessive, over dominant, and log-additive) were chosen to evaluate the associations between each SNP and gynecologic cancer risk. Statistical differences in SNPs’ genotype distribution were tested using a chi square test with estimation of the odds ratio (OR) for each genotype with respect to the reference genotype. Estimated ORs were obtained with 95% confidence intervals, and all statistical tests were considered bilateral with a significance level of 0.05. The best inheritance models were selected using the Akaike information criterion (AIC). Data analyses were performed using R software version 4.2.2 (R Foundation for Statistical Computing, Vienna, Austria, accessed on 26 January 2023, online: https://www.R-project.org/) [92] and the “SNPassoc” version 2.0.2 and “ggstatsplot” packages version 0.11.0 [93,94]. Interaction analyses were performed using the open source MDR software [95]. The statistical power was evaluated post hoc by the “genpwr” package version 1.0.2 in R [96], which analyzes the statistical power under the evaluation between true and test genetic models (Dominant, Additive, Recessive, 2 degree of freedom). As the sample comprised 440 subjects (200 with gynecologic cancers and 240 controls), and the MAF of selected SNPs ranged from 0.22 to 0.42, assuming a logistic model, case/control ratio = 200/440 = 0.455, alpha = 0.05, and with a power of 0.80, the smaller detectable odds ratio is 1.47.

## 5. Conclusions

Folic acid plays an important role in the synthesis and methylation of DNA and RNA; therefore, it is possible that its deficiency in the body contributes to genome instability and chromosome breaks, which are often responsible for the development of cancer. Both gynecologic cancer and depression are related to aging, with most cases being diagnosed in postmenopausal women, although family history and genetics play a role in the risk of these diseases. Cancers of the female reproductive system are strongly associated with the occurrence of depression, and ovarian cancer may be associated with the rs2236225 variant of the *MTHFD1* gene. In addition, in healthy aging women in the Polish population, the rs1801133 variant of the *MTHFR* gene is associated with depression. Treating depression, in addition to improving the quality of life, can also extend the survival of cancer patients.

### Strengths and Limitations

This research has several limitations. First, our study group is small and consists of patients with three types of gynecological cancers. We have not studied intermediates of the folic acid pathway, such as plasma tHcy. After menopause, the serum levels of homocysteine are higher than those in younger women. Some patients in the control group were taking substitutive hormone therapy, which can significantly reduce Hcy levels [97,98]. On the other hand, the treatment used in cancer patients could increase the risk of developing hyperhomocysteinemia [99]. We also did not include the effect of patients’ body weight in our analyses. Obesity is a strong risk factor for endometrial cancer, and the majority of these women were overweight or obese. Moreover, different types of cancer and the duration of treatment may have influenced the women’s body weight. It is also significant that we studied only three of the many polymorphic variants of various folate pathway genes. However, to our knowledge, this study is the first to assess SNP–SNP interactions in folate-related pathways associated with gynecologic cancers and depression. We were surprised by the small number of existing studies examining the association of polymorphic variants of the *MTR* and *MTHFD1* genes with gynecological malignancies. Thus, future large-scale studies are warranted in order to elucidate the association of gynecologic cancers with the single-carbon metabolic pathway. The strengths of this study are the group homogeneity and simultaneous study of women afflicted with cancer and depression.

## Figures and Tables

**Figure 1 ijms-24-12574-f001:**
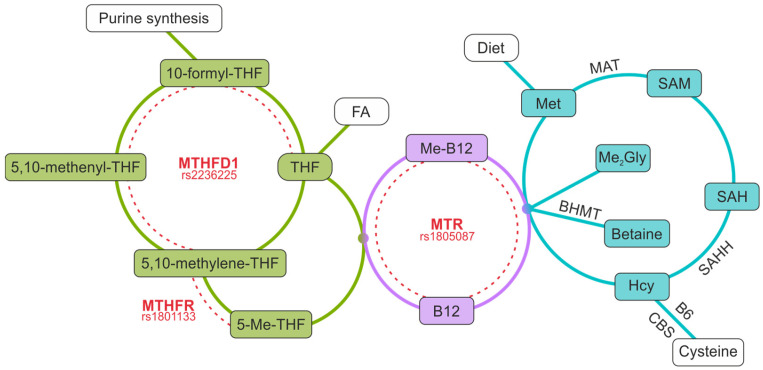
Schematic illustration of one-carbon metabolism pathways. The genes studied, their polymorphic variants, and the transformations in which they are involved (dotted line) are marked in red.

**Figure 2 ijms-24-12574-f002:**
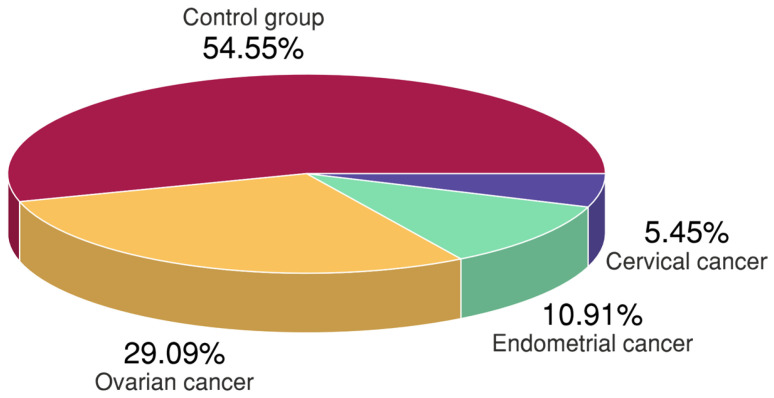
Pie chart showing the percentage of distribution of cases in the study groups.

**Figure 3 ijms-24-12574-f003:**
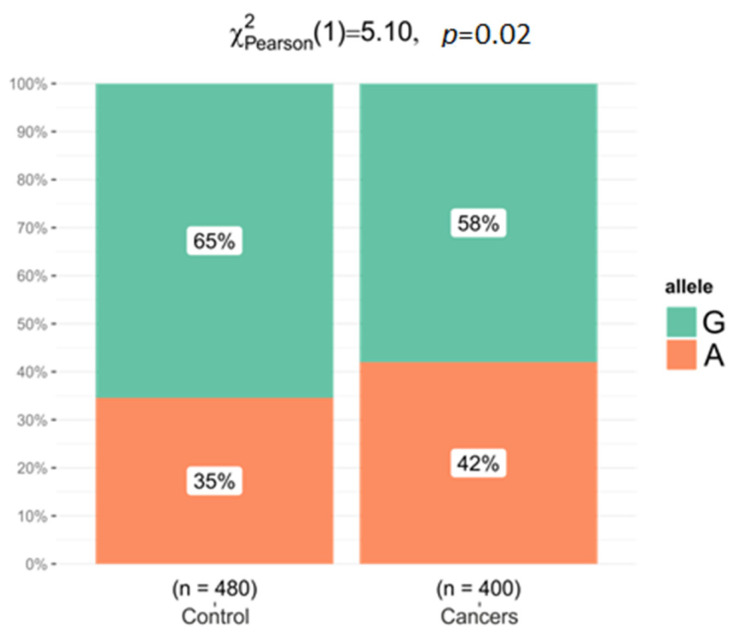
Observed allele frequencies for the *MTHFD1* gene rs2236225 variant in women with gynecologic cancers and controls.

**Figure 4 ijms-24-12574-f004:**
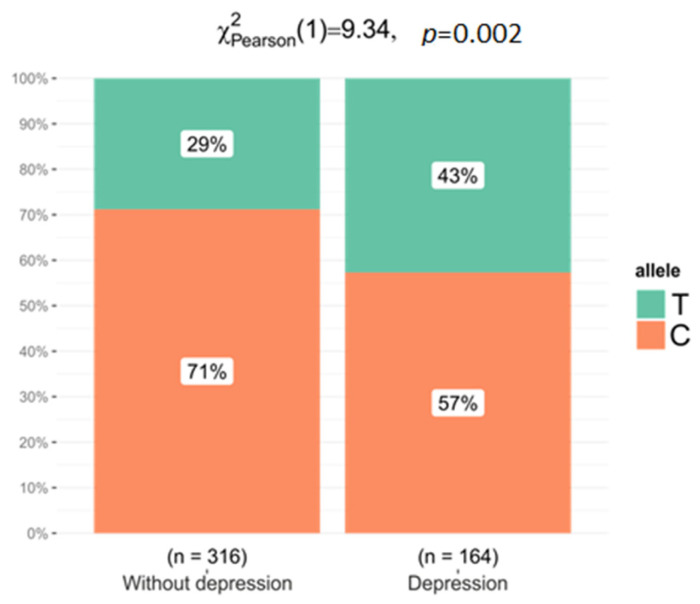
Observed allele frequencies for the *MTHFR* gene rs1801133 variant in control group with and without depression.

**Figure 5 ijms-24-12574-f005:**
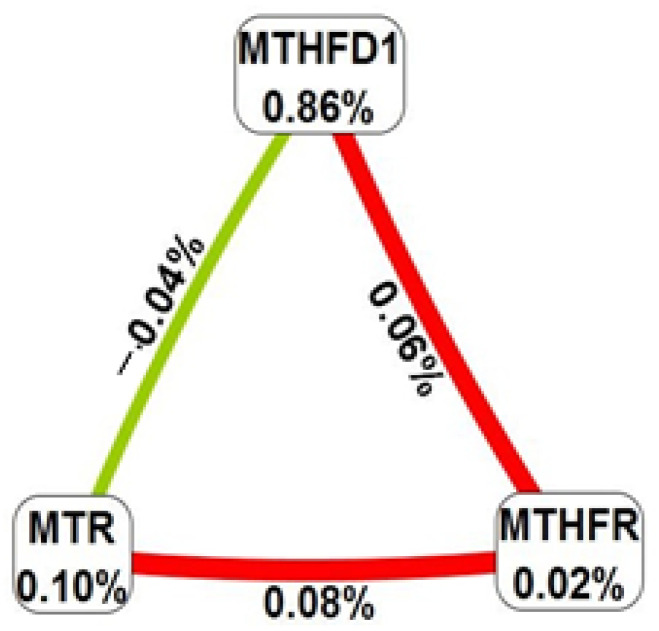
Fruchterman–Reingold graph for the interactions between SNPs. Positive-percent entropy indicates synergy (red lines), whereas negative-percent entropy indicates independence or redundancy (green line).

**Table 1 ijms-24-12574-t001:** Characteristics of the studied population.

Characteristics	Controls	Cancers	*p*
Sample size, *n*	240	200	
Age (years)	60.10 ± 7.82	61.40 ± 12.17	0.1920 *
Age at diagnosis (years)	-	59.59 ± 12.32	-
Postmenopausal status, *n* (%)			<0.001
Yes	164 (68.3)	198 (99.0)	
No	76 (31.7)	2 (1.0)	
Depression **, *n* (%)			<0.001
Yes	82 (34.2)	139 (69.5)	
No	158 (65.8)	61 (30.5)	
Depression **, median [IQR]			
Yes	12.0 [10.0–14.0]	13.0 [10.0–19.0]	0.008
No	4.0 [2.0–5.0]	4.0 [3.0–6.0]	0.249

mean ± SD, * Welch two-sample *t*-test. ** Depression was defined as a score greater than 7 on the HDRS.

**Table 2 ijms-24-12574-t002:** Depression levels of the patients estimated by the Hamilton Depression Rating Scale.

Depression Levels HDRS Points	Controls *n* = 240 (%)	Ovarian *n* = 128 (%)	Endometrial *n* = 48 (%)	Cervical *n* = 24 (%)	*p*
Not depressed: 0–7	158 (65.8%)	36 (28.1%)	16 (33.3%)	9 (37.5%)	<0.001
Mild (subthreshold): 8–13	55 (22.9%)	46 (35.9%)	19 (39.6%)	7 (29.2%)	
Moderate (mild): 14–18	21(8.8%)	18 (14.1%)	4 (8.3%)	3 (12.5%)	
Severe (moderate): 19–22	6 (2.5%)	12 (9.4%)	2 (4.2%)	3 (12.5%)	
Very severe (severe): >23	0 (0.0%)	16 (12.5%)	7 (14.6%)	2 (8.3%)	

*p*—categorical Pearson’s Chi-squared test.

**Table 3 ijms-24-12574-t003:** Prevalence of *MTHFR*, *MTR*, and *MTHFD1* alleles in cases (*n* = 400) and controls (*n* = 480).

SNP	Alleles	Controls *n* = 480		Cancers *n* = 400		*p*
		MAF *n* (Frequency)	HWE *p*	MAF *n* (Frequency)	HWE *p*	
*MTHFR* (rs1801133)	C>T	161 (0.335)	0.248	133 (0.332)	0.114	0.927
*MTR* (rs1805087)	A>G	106 (0.221)	0.452	94 (0.235)	1.000	0.618
*MTHFD1* (rs2236225)	G>A	166 (0.346)	1.000	168 (0.420)	0.773	0.024

SNP—single-nucleotide polymorphism; MAF—minor allele frequency; HWE—Hardy-Weinberg Equilibrium; *p*—Pearson’s chi-squared test.

**Table 4 ijms-24-12574-t004:** Distribution of genotype frequencies of *MTHFR*, *MTR*, and *MTHFD1* polymorphisms in women with cancers and control subjects.

SNP/Genetic Model	Genotypes	Controls *n* (%)	Cancers *n* (%)	OR (95% CI)	*p*	AIC
	*MTHFR* (rs1801133)					
Codominant	CC	110 (45.8)	94 (47.0)	1.00	0.931	612.2
	CT	99 (41.2)	79 (39.5)	0.93 (0.62–1.40)		
	TT	31 (12.9)	27 (13.5)	1.02 (0.57–1.83)		
Dominant	CC vs. CT-TT	130 (54.2)	106 (53.0)	0.95 (0.65–1.39)	0.807	610.3
Recessive	CC-CT vs. TT	209 (87.1)	173 (86.5)	1.05 (0.60–1.83)	0.857	610.3
Over dominant	CC-TT vs. CT	141 (58.8)	121 (60.5)	0.93 (0.63–1.36)	0.710	610.2
log-Additive	1, 2, 3	240 (54.5)	200 (45.5)	0.99 (0.75–1.29)	0.930	610.3
	*MTR* (rs1805087)					
Codominant	AA	148 (61.7)	117 (58.5)	1.00	0.743	611.7
	AG	78 (32.5)	72 (36.0)	1.17 (0.78–1.74)		
	GG	14 (5.8)	11 (5.5)	0.99 (0.44–2.27)		
Dominant	AA vs. AG-GG	92 (38.3)	83 (41.5)	1.14 (0.78–1.67)	0.499	609.9
Recessive	AA-AG vs. GG	226 (94.2)	189 (94.5)	0.94 (0.42–2.12)	0.880	610.3
Over dominant	AA-GG vs. AG	162 (67.5)	128 (64.0)	1.17 (0.79–1.73)	0.441	609.7
log-Additive	1, 2, 3	240 (54.5)	200 (45.5)	1.08 (0.79–1.48)	0.623	610.1
	*MTHFD1* (rs2236225)					
Codominant	GG	103 (42.9)	66 (33.0)	1.00	0.073	607.1
	GA	108 (45.0)	100 (50.0)	1.45 (0.96–2.18)		
	AA	29 (12.1)	34 (17.0)	1.83 (1.02–3.28)		
Dominant	GG vs. GA-AA	137 (57.1)	134 (67.0)	1.53 (1.03–2.25)	0.033	605.8
Recessive	GG-GA vs. AA	211 (87.9)	166 (83.0)	1.49 (0.87–2.55)	0.144	608.2
Over dominant	GG-AA vs. GA	132 (55.0)	100 (50.0)	1.22 (0.84–1.78)	0.300	609.2
log-Additive	1, 2, 3	240 (54.5)	200 (45.5)	1.37 (1.04–1.81)	0.024	605.2

*n*—number; OR—odds ratio; 95% CI—confidence interval; AIC—Akaike information criterion.

**Table 5 ijms-24-12574-t005:** Distribution of genotype frequencies of studied polymorphisms in women with cancer divided by cancer category.

SNP/Genotypes		Ovarian *n* = 128 (%)	Endometrial *n* = 48 (%)	Cervical *n* = 24 (%)	*p*
*MTHFR* (rs1801133)	CC	59 (46.1)	23 (47.9)	12 (50.0)	0.835
	CT	49 (38.3)	20 (41.7)	10 (41.7)	
	TT	20 (15.6)	5 (10.4)	2 (8.3)	
*MTR* (rs1805087)	AA	76 (59.4)	28 (58.3)	13 (54.2)	0.977
	AG	45 (35.2)	17 (35.4)	10 (41.7)	
	GG	7 (5.5)	3 (6.2)	1 (4.2)	
*MTHFD1* (rs2236225)	GG	38 (29.7)	20 (41.7)	8 (33.3)	0.441
	GA	68 (53.1)	22 (45.8)	10 (41.7)	
	AA	22 (17.2)	6 (12.5)	6 (25.0)	

*p*—categorical Pearson’s chi-squared test.

**Table 6 ijms-24-12574-t006:** Distribution of genotype frequencies of studied polymorphisms in women with cancers and control subjects divided according to depression status.

SNP/Genotypes		Cancers		*p*	Controls		*p*
		Depressed *n* = 139 (%)	Not Depressed *n* = 61 (%)		Depressed *n* = 82 (%)	Not Depressed *n* = 158 (%)	
*MTHFR* (rs1801133)	CC	64 (46.0)	30 (49.2)	0.919	29 (35.4)	81 (51.3)	0.011
	CT	56 (40.3)	23 (37.7)		36 (43.9)	63 (39.9)	
	TT	19 (13.7)	8 (13.1)		17 (20.7)	14 (8.8)	
*MTR* (rs1805087)	AA	80 (57.6)	37 (60.7)	0.912	50 (61.0)	98 (62.0)	0.984
	AG	51 (36.7)	21 (34.4)		27 (32.9)	51 (32.3)	
	GG	8 (5.8)	3 (4.9)		5 (6.1)	9 (5.7)	
*MTHFD1* (rs2236225)	GG	47 (33.8)	19 (31.1)	0.897	39 (47.6)	64 (40.5)	0.376
	GA	68 (48.9)	32 (52.5)		36 (43.9)	72 (45.6)	
	AA	24 (17.3)	10 (16.4)		7 (8.5)	22 (13.9)	

*p*—categorical Pearson’s chi-squared test.

**Table 7 ijms-24-12574-t007:** Gene-gene interaction analysis between gynecologic cancer patients and controls.

Model	Training Balanced Accuracy	Testing Balanced Accuracy	CVC	OR (95% CI)	*p*
*MTHFD1*	0.5496	0.5496	10/10	1.53 (1.03–2.25)	0.033
*MTHFR*, *MTHFD1*	0.5588	0.5208	8/10	1.59 (1.09–2.34)	0.016
*MTHFR*, *MTR*, *MTHFD1*	0.5787	0.5221	10/10	1.99 (1.33–2.98)	0.001

*p* values were calculated using χ^2^ test; CVC—cross-validation consistency.

**Table 8 ijms-24-12574-t008:** Primary information of the selected genes and variants.

Gene Symbol	rs No.	Location *	Alleles	MAF **
*MTHFR*	rs1801133	chr1:11796321	C>T	T = 0.3648
*MTR*	rs1805087	chr1:236885200	A>G	G = 0.1730
*MTHFD1*	rs2236225	chr14:64442127	G>A	A = 0.4294

* Location on chromosome based on human reference sequence (GRCh38.p13). ** MAF—minor allele frequency (1000 Genomes Project, EUR samples).

## Data Availability

Not applicable.

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
