# Peer review of "Common Variants in One-Carbon Metabolism Genes (MTHFR, MTR, MTHFD1) and Depression in Gynecologic Cancers"

_ijms, 2023, doi:10.3390/ijms241612574_

Round 1
Reviewer 1 Report
The authors investigated the correlations between one-carbon metabolism and gynecological risks. Interactions between variants involved in one-carbon metabolism and depression were also discussed.
Major issues:
1, In the introduction part, it will be better if you can add some previous research about relationships between depression and gynecologic malignancy.
2, You lack a validation cohort to validate the results
Minor issues:
1, You present many data using tables but it will be better if you can visualize them using colorful figures, which will be more straigtforward.
Author Response
Dear Reviewer,
We would like to submit a revised version of our manuscript. We appreciate the overall positive evaluation of our manuscript by the reviewers and we would like to thank the reviewers for their efforts and constructive criticism that helped us to improve the quality of our manuscript. Please find below a point by point response to the issues raised by the reviewer.
All changes have been placed in corrected manuscript in green.
Reviewer 1
Major issues:
- In the introduction part, it will be better if you can add some previous research about relationship between depression and gynecologic malignancy.
Answer: Systematic review of the literature on the incidence of depression and anxiety in patients with ovarian cancer (sample of 3623 patients) by stage of treatment. suggests that the prevalence of depression and anxiety in these women, across the treatment spectrum, is significantly higher than in the population of healthy women [Watts et al., 2015]. Effective screening and treatment of depression symptoms may have important oncological consequences In meta-analysis Walker et al. (2021) examined whether depression is associated with worse survival in people with cancer. They analyzed data on 20,582 patients with breast, colorectal, gynecological, lung, and prostate cancers from Scotland, United Kingdom. Major depression was associated with worse survival for all cancers types. For gynecological cancers hazard ratio was 1.36 (95% CI = 1.08-1.71) [Walker et al.; 2021].
Watts S, Prescott P, Mason J, McLeod N, Lewith G. Depression and anxiety in ovarian cancer: a systematic review and meta-analysis of prevalence rates. BMJ Open. 2015 Nov 30;5(11):e007618. doi: 10.1136/bmjopen-2015-007618. PMID: 26621509; PMCID: PMC4679843.
Walker J, Mulick A, Magill N, Symeonides S, Gourley C, Burke K, Belot A, Quartagno M, van Niekerk M, Toynbee M, Frost C, Sharpe M. Major Depression and Survival in People With Cancer. Psychosom Med. 2021 Jun 1;83(5):410-416. doi: 10.1097/PSY.0000000000000942. PMID: 33938501.
- You lack a validation cohort to validate the results.
Answer: The statistical power was evaluated post-hoc by the "genpwr" package version 1.0.2 in R [Moore et al.,2020, accessed on 21 June 2023], which analyzes the statistical power under the evaluation between true and test genetic models (Dominant, Additive, Recessive, 2 degree of freedom). As the sample comprised 440 subjects (200 with gynecologic cancers and 240 controls) and the MAF of selected SNPs ranged from 0.22 to 0.42, assuming a logistic model, case/control ratio = 200/440 = 0.455, alpha = 0.05 and a power of 0.80, the smaller detectable odds ratio is 1.47.
Moore CM, Jacobson SA, Fingerlin TE. Power and Sample Size Calculations for Genetic Association Studies in the Presence of Genetic Model Misspecification. Hum Hered. 2019;84(6):256-271. doi: 10.1159/000508558. Epub 2020 Jul 28. PMID: 32721961; PMCID: PMC7666027.
Minor issues:
- You present many data using tables but it will be better if you can visualize them using colorful figures, which will be more straigforward.
Answer: A Fig.2 showing the percentage of distribution of cases in the study groups has been added.
New 21 references have been added.

Reviewer 2 Report
Abstract: We investigated the association between methylenetetrahydrofolate reductase (MTHFR 21 677C>T, rs1801133), 5-methyltetrahydrofolate-homocysteine methyltransferase (MTR 2756A>G, 22 rs1805087), and methylenetetrahydrofolate dehydrogenase, cyclohydrolase and formyltetrahy-23 drofolate synthetase 1 (MTHFD1 1958G>A, rs2236225)—well-studied functional variants involved 24 in one-carbon metabolism—and gynecologic cancer risk, and the interaction between these poly-25 morphisms and depression. A total of 200 gynecologic cancer cases and 240 healthy controls were 26 recruited to participate in this study. Three single nucleotide variants (SNVs) (rs1801133, rs1805087, 27 rs2236225) were genotyped using the PCR-restriction fragment length polymorphism method. 28 Depression was assessed in all patients using the Hamilton Depression Scale. Depression was 29 statistically significantly more frequent in women with gynecologic cancers (69.5% vs. 34.2% in 30 controls, p<0.001). MTHFD1 rs2236225 was associated with an increased risk of gynecologic can-31 cers (in dominant OR=1.53, p=0.033, and in log-additive models OR=1.37, p=0.024). Also an associ-32 ation was found between depression risk and MTHFR rs1801133 genotypes in the controls but not 33 in women with gynecologic cancers (in codominant model CC vs. TT: OR=3.39, 95%: 1.49–7.74, 34 p=0.011). Cancers of the female reproductive system are associated with the occurrence of depres-35 sion, and ovarian cancer may be associated with the rs2236225 variant of the MTHFD1 gene. In 36 addition, in healthy aging women in the Polish population, the rs1801133 variant of the MTHFR 37 gene is associated with depression.
Q1. Figure 1 is a helpful schematic illustration that outlines the intricate pathways involved in one-carbon metabolism, a vital biochemical process that influences DNA synthesis, methylation reactions, and nucleotide synthesis. This diagram provides a clear overview of the key enzymes and molecules participating in this pathway, aiding in the understanding of its complexity.
In this figure, the genes under investigation in the study are distinguished by being highlighted in red. Specifically, the genes studied are methylenetetrahydrofolate reductase (MTHFR), 5-methyltetrahydrofolate-homocysteine methyltransferase (MTR), and methylenetetrahydrofolate dehydrogenase, cyclohydrolase, and formyltetrahydrofolate synthetase 1 (MTHFD1). These genes possess various functional variants, and the research aims to explore the potential association between these specific polymorphic variants and the risk of gynecologic cancer.
By visually indicating the genes and their corresponding polymorphic variants in the diagram, Figure 1 effectively presents the specific genetic elements that are the focal point of the study. This visual representation helps elucidate the genetic context and emphasizes the significance of these genes and their variants within the broader context of the one-carbon metabolism pathway. Suggest revising the figure 1 more clearly.
Q2. What are the depression levels of the patients, as estimated by the Hamilton Depression Rating Scale, according to Table 2?
Q3. What is the genotype distribution of the SNPs rs1801133, rs1805087, and rs2236225 in the MTHFR, MTR, and MTHFD1 genes in the study participants (cases and controls)?
Q4. Are the genotype distributions of these three polymorphisms consistent with the Hardy-Weinberg equilibrium in both cases and controls?
Q5. What are the minor allele frequencies (MAFs) of the MTHFR, MTR, and MTHFD1 alleles in cases and controls?
Q.6Is there a significant difference in the frequency of the A allele in the rs2236225 locus of the MTHFD1 gene between women with gynecologic cancers and controls?
Q7. What are the genotype frequencies of the MTHFR, MTR, and MTHFD1 polymorphisms in women with gynecologic cancers and control subjects?
Q8. Is there an association between the genotypes of rs1801133, rs1805087, and rs2236225 and the risk of gynecologic cancers?
Q9. What is the sample size of women with cancer in the study?
Q10. How many women were included in each cancer category (ovarian, endometrial, cervical)?
Q11.What is the distribution of MTHFR (rs1801133) genotypes in women with ovarian cancer?
Q12.What is the distribution of MTHFR (rs1801133) genotypes in women with endometrial cancer?
Q13.What is the distribution of MTHFR (rs1801133) genotypes in women with cervical cancer?
Q14.Is there a statistically significant difference in the MTHFR (rs1801133) genotype distribution between ovarian, endometrial, and cervical cancer?
Q15.What is the distribution of MTR (rs1805087) genotypes in women with ovarian cancer?
Q16.What is the distribution of MTR (rs1805087) genotypes in women with endometrial cancer?
Q17.What is the distribution of MTR (rs1805087) genotypes in women with cervical cancer?
Q18.Is there a statistically significant difference in the MTR (rs1805087) genotype distribution between ovarian, endometrial, and cervical cancer?
Q19.What is the distribution of MTHFD1 (rs2236225) genotypes in women with ovarian cancer?
Q20.What is the distribution of MTHFD1 (rs2236225) genotypes in women with endometrial cancer?
Q21.What is the distribution of MTHFD1 (rs2236225) genotypes in women with cervical cancer?
Q22.Is there a statistically significant difference in the MTHFD1 (rs2236225) genotype distribution between ovarian, endometrial, and cervical cancer?
Q23.Are there any statistically significant associations between the studied polymorphisms and the occurrence of depression?
Q24.What is the association between the MTHFR (rs1801133) genotype and depression risk in the control group?
Q25.Is there an association between depression risk and MTHFR (rs1801133) genotypes in women with gynecologic cancers?
Q26.What is the distribution of genotypes of studied polymorphisms in women with cancers and control subjects divided according to depression status?
Q27.What is the gene-gene interaction analysis between gynecologic cancer patients and controls?
Q28.Which gene-gene interaction model had the highest testing accuracy and cross-validation consistency?
Additional comments
EX1. Clarity and Structure: The data presented in Table 5 and Table 6 provide a clear overview of the genotype frequencies among women with different types of cancer and controls, as well as their division based on depression status. The tables are well-organized and easy to understand, facilitating the interpretation of the results.
EX2. Statistical Analysis: The statistical analysis conducted using categorical Pearson's chi-squared test is appropriate for examining the associations between genotypes and different variables. The p-values provided offer insights into the significance of the observed differences. It would be helpful to include the sample sizes (n) for each group to better understand the statistical power of the study.
EX3. Associations with Cancer Types: The results indicate that statistically significant differences were observed only between ovarian cancer and the control group for certain genotypes. It would be valuable to discuss the potential implications of these findings in the context of ovarian cancer development and its genetic predisposition.
EX4. Association with Depression: The association between the MTHFR gene variant (rs1801133) and the occurrence of depression is an interesting finding. It is noteworthy that this association was observed in the control group but not in women with gynecologic cancers. Exploring the possible mechanisms underlying this association could provide further insights into the interplay between genetic factors and mental health outcomes.
EX5. SNP-SNP Interaction: The utilization of multifactor dimensionality reduction (MDR) analysis to explore gene-gene interactions is commendable. The identification of the best single-locus and two-locus models provides a basis for understanding the combined effects of multiple SNPs in predicting gynecologic cancers. However, further discussion on the biological plausibility and potential functional implications of these interactions would enhance the interpretation of the results.
EX6. Discussion: The discussion section provides a comprehensive overview of the current understanding of depression among cancer patients and the role of homocysteine metabolism in both depression and cancer. The inclusion of previous research findings and meta-analyses strengthens the context of your study. However, additional discussion on the potential clinical implications and future research directions based on your results would be beneficial.
Overall, the research contributes to the existing knowledge on the genetic associations with gynecologic cancers and their link to depression. The clear presentation of data and appropriate statistical analysis support the validity of the findings. Addressing the above points in the presentation will further enhance the understanding and impact of this research.
This study presentation is generally clear and concise, which is essential for effective communication. However, in some sections, the authors could simplify or rephrase sentences to enhance clarity. Pay attention to the use of complex sentence structures or jargon that may be challenging for the audience to understand. Strive for simplicity and clarity in your language to ensure that the message is easily comprehensible.
Author Response
Dear Reviewer,
We would like to submit a revised version of our manuscript. We appreciate the overall positive evaluation of our manuscript by the reviewers and we would like to thank the reviewers for their efforts and constructive criticism that helped us to improve the quality of our manuscript. Please find below a point by point response to the issues raised by the reviewer.
All changes have been placed in corrected manuscript in green.
Reviewer 2
Q1: Suggest revising the figure 1 more clearly.
Answer:
Fig. 1. Schematic illustration of one-carbon metabolism pathways. The genes studied, their polymorphic variants and transformations in which they are involved (dotted line) are marked in red.
Q2: What are the depression levels of the patients, as estimated by the Hamilton Depression Rating Scale, according to Table 2?
Answer:
The median (IQR) of the Hamilton Depression Rating Scale in women with depression was statistically significantly higher in the group with cancer and was 13 [10 - 19] vs. 12 [10 - 14] in the control group (p=0.008)
Table 1.
|
Depression**, median [IQR] Yes No |
12.0 [10.0-14.0] 4.0 [ 2.0- 5.0] |
13.0 [10.0-19.0] 4.0 [ 3.0- 6.0] |
0.008 0.249 |
Q3: What is the genotype distribution of the SNPs rs1801133, rs1805087, and rs2236225 in the MTHFR, MTR and MTHFD1 genes in the study participants (cases and controls)?
Answer: The genotype distribution of the rs1801133, rs1805087 and rs2236225 variants for controls and cases is shown in Table 4.
Q4: Are the genotype distributions of these three polymorphisms consistent with the Hardy-Weinberg equilibrium in both cases and controls?
Answer: Genotype distribution of these three polymorphisms in cases and controls was consistent with the Hardy–Weinberg equilibrium. P-values presented in Table 3.
Q5: What are the minor allele frequencies (MAFs) of the MTHFR, MTR and MTHFD1 alleles in cases and controls?
Answer: Minor allele frequencies for cases and controls are shown in Table 3.
Q6: Is there a significant difference in the frequency of the A allele in the rs2236225 locus of the MTHFD1 gene between women with gynecologic cancers and controls?
Answer: There is a significant difference in the frequency of the A allele in the rs2236225 locus of the MTHFD1 gene between women with gynecologic cancers and controls.
“Allele A MTHFD1 rs2236225 was more common in cancer patients than in controls (42.0% vs. 34.6%, OR=0.730, 95% CI: 0.55– 0.96, p=0.024) (Table 3, Fig. 3).”
Q7: What are the genotype frequencies of the MTHFR, MTR and MTHFD1 polymorphisms in women with gynecologic cancers and controls?
Answer: Genotype frequencies of the MTHFR, MTR and MTHFD1 polymorphisms in women with gynecologic cancers and controls are shown in Table 4.
The genotype frequency distribution of rs1801133, rs1805087 and rs2236225 and their association with gynecologic cancers risk are shown in Table 4.
Q8: Is there an association between the genotypes of rs1801133, rs1805087 and rs2236225 and the risk of gynecologic cancers?
Answer: The genotype frequency distribution of rs1801133, rs1805087 and rs2236225 and their association with gynecologic cancers risk are shown in Table 4.
The MTHFD1 rs2236225 AA genotype was associated with an increased risk of cancers (GG vs. AA, OR=1.83, 95%: 1.02–3.28, p=0.073; in dominant model GG vs. GA-AA, OR 1.53, 95%: 1.03–2.25, p=0.033, AIC=605.8 and in log-additive model OR=1.37, 95%: 1.04–1.81, p=0.024, AIC=605.2).
No statistically significant associations were observed between MTHFR rs180133 and MTR rs1805087 variants and the risk of cancers.
Q9: What is the sample size of women with cancer in the study?
Answer: A total of 200 unrelated women with gynecologic cancers ranging in age from 18 to 90 years (mean ± SD; 60.40 ± 12.17) and 240 controls ranging from 43 to 75 years old (mean ± SD; 60.10 ± 7.82) were enrolled in the hospital-based case-control study.
Q9: How many women were included in each cancer category (ovarian, endometrial, cervical)?
Answer: In the gynecologic cancers group 128 women had ovarian cancer, 48 endometrial cancer, and 24 cervical cancer. Figure 2 showing the percentage distribution of the study population.
Fig. 2. Pie chart showing the percentage of distribution of cases in the study groups.
Q11: What is the distribution of MTHFR (rs18011330) genotypes in women with ovarian cancer?
Answer: Table 5.
Q12: What is the distribution of MTHFR (rs18011330) genotypes in women with endometrial cancer?
Answer: Table 5.
Q13: What is the distribution of MTHFR (rs18011330) genotypes in women with cervical cancer?
Answer: Table 5. Distribution of genotype frequencies of studied polymorphisms in women with cancer divided by cancer category
|
SNP/genotypes |
Ovarian n=128 (%) |
Endometrial n=48 (%) |
Cervical n=24 (%) |
p |
|
|
MTHFR (rs1801133) |
CC |
59 (46.1) |
23 (47.9) |
12 (50.0) |
0.835 |
|
CT |
49 (38.3) |
20 (41.7) |
10 (41.7) |
||
|
TT |
20 (15.6) |
5 (10.4) |
2 ( 8.3) |
||
Q14: Is there a statistically significant difference in the MTHFR (rs18011330) genotype distribution between ovarian, endometrial and cervical cancer?
Answer: There is no statistically significant difference in the distribution of the MTHFR (rs18011330) genotype between ovarian, endometrial and cervical cancer, p=0.835 (Table 5)
Q15: What is the distribution of MTR (rs180587) genotypes in women with ovarian cancer?
Table 5.
Q16: What is the distribution of MTR (rs180587) genotypes in women with endometrial cancer?
Table 5.
Q17: What is the distribution of MTR (rs180587) genotypes in women with cervical cancer?
Answer: Table 5. Distribution of genotype frequencies of studied polymorphisms in women with cancer divided by cancer category
|
SNP/genotypes |
Ovarian n=128 (%) |
Endometrial n=48 (%) |
Cervical n=24 (%) |
p |
|
|
MTR (rs1805087) |
AA |
76 (59.4) |
28 (58.3) |
13 (54.2) |
0.977 |
|
AG |
45 (35.2) |
17 (35.4) |
10 (41.7) |
||
|
GG |
7 (5.5) |
3 (6.2) |
1 (4.2) |
||
Q18: Is there a statistically significant difference in the MTR (rs180587) genotype distribution between ovarian, endometrial and cervical cancer?
Answer: There is no statistically significant difference in the distribution of the MTR (rs180587) genotype between ovarian, endometrial and cervical cancer, p=0.977 (Table 5).
Q19: What is the distribution of MTHFD1 (rs2236225) genotypes in women with ovarian cancer?
Table 5.
Q20: What is the distribution of MTHFD1 (rs2236225) genotypes in women with endometrial cancer?
Table 5.
Q21: What is the distribution of MTHFD1 (rs2236225) genotypes in women with cervical cancer?
Table 5. Distribution of genotype frequencies of studied polymorphisms in women with cancer divided by cancer category
|
SNP/genotypes |
Ovarian n=128 (%) |
Endometrial n=48 (%) |
Cervical n=24 (%) |
p |
|
|
MTHFD1 (rs2236225) |
GG |
38 (29.7) |
20 (41.7) |
8 (33.3) |
0.441 |
|
GA |
68 (53.1) |
22 (45.8) |
10 (41.7) |
||
|
AA |
22 (17.2) |
6 (12.5) |
6 (25.0) |
||
Q22: Is there a statistically significant difference in the MTHFD1 (rs2236225) genotype distribution between ovarian, endometrial and cervical cancer?
Answer: There is no statistically significant difference in the distribution of the MTHFD1 (rs2236225) genotype between ovarian, endometrial and cervical cancer, p=0.441 (Table 5)
Q23: Are there any statistically significant associations between the studied polymorphisms and occurrence of depression?
Answer: There is no statistically significant difference in the associations between the studied polymorphisms and occurrence of depression.
The comparison of the frequency of genotypes between women with depression (n=221) and women without depression (n=219) indicated only the possibleassociation of the rs1801133 variant of the MTHFR gene with the occurrence of depression (p=0.076 in codominant model).
Q24: What is the associations between the MTHFR (rs18011330) genotype and depression risk in the control group?
Q25: Is there an associations between depression risk and MTHFR (rs18011330) genotypes in women with gynecologic cancers?
Answer: We conducted stratification analysis by depression status in case and control groups (Table 6), and it revealed an association between depression risk and MTHFR rs1801133 genotypes in the controls (in codominant model CC vs. TT: OR=3.39, 95%: 1.49–7.74, p=0.011), but not in women with gynecologic cancers.
Q26: What is the distribution of genotypes of studied polymorphisms in women with cancers and control subjects divided according to depression status?
Answer: The genotype distribution of studied polymorphisms in women with cancers and control subjects divided according to depression status is presented in Table 6.
Table 6. Distribution of genotype frequencies of studied polymorphisms in women with cancers and control subjects divided according to depression status.
|
SNP/genotypes |
Cancers |
p |
Controls |
p |
|||
|
Depressed n=139 (%) |
Not depressed n=61 (%) |
Depressed n=82 (%) |
Not depressed n=158 (%) |
||||
|
MTHFR (rs1801133) |
CC |
64 (46.0) |
30 (49.2) |
0.919 |
29 (35.4) |
81 (51.3) |
0.011 |
|
CT |
56 (40.3) |
23 (37.7) |
36 (43.9) |
63 (39.9) |
|||
|
TT |
19 (13.7) |
8 (13.1) |
17 (20.7) |
14 (8.8) |
|||
|
MTR (rs1805087) |
AA |
80 (57.6) |
37 (60.7) |
0.912 |
50 (61.0) |
98 (62.0) |
0.984 |
|
AG |
51 (36.7) |
21 (34.4) |
27 (32.9) |
51 (32.3) |
|||
|
GG |
8 (5.8) |
3 (4.9) |
5 (6.1) |
9 (5.7) |
|||
|
MTHFD1 (rs2236225) |
GG |
47 (33.8) |
19 (31.1) |
0.897 |
39 (47.6) |
64 (40.5) |
0.376 |
|
GA |
68 (48.9) |
32 (52.5) |
36 (43.9) |
72 (45.6) |
|||
|
AA |
24 (17.3) |
10 (16.4) |
7 (8.5) |
22 (13.9) |
|||
p – categorical Pearson's chi-squared test
Q27: What is the gene-gene interaction analysis between gynecologic cancer patients and controls?
Answer: Performed using the MDR software gene-gene interaction analysis between gynecologic cancer patients and control is presented in Table 7.
Q28: Which gene-gene interaction model had the highest testing accuracy and cross-validation consistency?
The highest testing accuracy and cross-validation consistency for single-locus, two- and three- locus models to predict gynecologic cancers is presented in table 8.
The best single-locus model to predict gynecologic cancers was rs2236225 (testing accuracy, 0.5496; p=0.033; cross-validation consistency, 10/10). The best two-locus model was a combination of rs1801133 and rs2236225 with the testing accuracy of 0.5208 and cross-validation consistency of 8/10.
Additional comments
Ex1. Clarity and structure: The data presented in Table 5 and Table 6 provide a clear overview of the genotype frequencies among women with different types of cancer and controls as well as their division based on depression status. The tables are well-organized and easy to understand, facilitating the interpretation of the results.
Ex2. Statistical Analysis: The statistical analysis conducted using categorical Person’s chi-squared test is appropriate for examining the associations between genotypes and different variables. The p-values provided offer insights into the significance of the observed differences. It would be helpful to include the sample sizes (n) for each group to better understand the statistical power of the study.
Answer: We added in Statistical Analysis:
The statistical power was evaluated post-hoc by the "genpwr" package version 1.0.2 in R [Moore et al.,2020, accessed on 21 June 2023], which analyzes the statistical power under the evaluation between true and test genetic models (Dominant, Additive, Recessive, 2 degree of freedom). As the sample comprised 440 subjects (200 with gynecologic cancers and 240 controls) and the MAF of selected SNPs ranged from 0.22 to 0.42, assuming a logistic model, case/control ratio = 200/440 = 0.455, alpha = 0.05 and a power of 0.80, the smaller detectable odds ratio is 1.47.
Ex3. Associations with Cancer Types: The results indicate that statistically significant differences were observed only between ovarian cancer and the control group for certain genotypes. It would be valuable to discuss the potential implications of these findings in the context of ovarian cancer development and its genetic predisposition.
Answer: A genetic predisposition to ovarian cancer is suspected. Risk of getting sick increased in women with a family history of breast, uterine, or ovarian cancer in a mother or sister [Mori et al., 1988]. Many studies focus on explaining the relationship between polymorphisms related to folate metabolism and the increased risk of ovarian cancer, but the presented results remain controversial [Karimi-Zarchi et al., 2019, Tanha et al., 2021]. A cytoplasmic trifunctional enzyme MTHFD1 catalyzes reactions in the pathway of conversion of tetrahydrofolate (THF), the active form of folate, into substrates essential for the de novo purine and thymidylate synthesis. Cancer cells rely on nucleotide metabolism, and the folate cycle is a necessary pathway for tumor growth. Also, the MTHFD1 indirectly provides one-carbon units for methylation reactions by the synthesis of 5,10 methylene-THF. The 1958G>A MTHFD1 variant may influence DNA synthesis reactions and cell development, eventually affecting carcinogenesis. This variant is in the MTHFD1 synthetase domain, which produces 10-formyltetrahydrofolate (formylTHF) for the synthesis of purines from formate and THF. The 1958G>A polymorphism have been studied as associated to cancer disease, although with not univocal results. Wang and colleagues showed that the polymorphic 1958AA variant was linked to a significantly increased risk for gastric cancer as compared with the 1958GG or 1958AG genotypes [Wang et al., 2007], whereas Moruzzi showed that the 1958AA genotype was significantly less frequent among cancer patients and related to 75% reduction of colon cancer risk [Moruzzi et al., 2017]. The 1958AA increases the thermostability of the protein, which changes its metabolic activity and impairs the synthesis of purines, which may hinder tumor growth and progression and have a protective effect in cancer. Also the increased production of formylTHF by the MTHFD1 synthetase may increases the expression of the mitochondrial enzyme MTHFD2 [Moruzzi et al., 2017, Lévesque et al., 2017]. MTHFD2 is expressed in the developing embryo and is absent in most adult tissues [Ben-Sahra et al., 2016, Ju et al., 2019]. Interestingly, markedly elevated expression of MTHFD2 was identified in many cancers and correlates with poor survival in breast cancer patients [Liu et al., 2014, Nilsson et al., 2014]. Recently the role of MTHFD2 in ovarian cancer was investigated by Cui et al. (2022). This study found that MTHFD2 is highly expressed in ovarian cancer, as well as an indispensable risk factor for poor prognosis in patients with ovarian cancer. Also has a role in malignancy mainly through the MOB1A signaling, which is a potential target for treating ovarian cancer [Cui et al., 2022].
Mori M, Harabuchi I, Miyake H, Casagrande JT, Henderson BE, Ross RK. Reproductive, genetic, and dietary risk factors for ovarian cancer. Am J Epidemiol. 1988 Oct;128(4):771-7. doi: 10.1093/oxfordjournals.aje.a115030. PMID: 3421242.
Karimi-Zarchi M, Moghimi M, Abbasi H, Hadadan A, Salimi E, Morovati-Sharifabad M, Akbarian-Bafghi MJ, Zare-Shehneh M, Mosavi-Jarrahi A, Neamatzadeh H. Association of MTHFR 677C>T Polymorphism with Susceptibility to Ovarian and Cervical Cancers: A Systematic Review and Meta-Analysis. Asian Pac J Cancer Prev. 2019 Sep 1;20(9):2569-2577. doi: 10.31557/APJCP.2019.20.9.2569.
Tanha K, Mottaghi A, Nojomi M, Moradi M, Rajabzadeh R, Lotfi S, Janani L. Investigation on factors associated with ovarian cancer: an umbrella review of systematic review and meta-analyses. J Ovarian Res. 2021 Nov 11;14(1):153. doi: 10.1186/s13048-021-00911-z.
Wang L, Ke Q, Chen W, Wang J, Tan Y, Zhou Y, Hua Z, Ding W, Niu J, Shen J, Zhang Z, Wang X, Xu Y, Shen H. Polymorphisms of MTHFD, plasma homocysteine levels, and risk of gastric cancer in a high-risk Chinese population. Clin Cancer Res. 2007 Apr 15;13(8):2526-32. doi: 10.1158/1078-0432.CCR-06-2293. PMID: 17438114.
Moruzzi S, Guarini P, Udali S, Ruzzenente A, Guglielmi A, Conci S, Pattini P, Martinelli N, Olivieri O, Tammen SA, Choi SW, Friso S. One-carbon genetic variants and the role of MTHFD1 1958G>A in liver and colon cancer risk according to global DNA methylation. PLoS One. 2017 Oct 2;12(10):e0185792. doi: 10.1371/journal.pone.0185792. PMID: 28968444; PMCID: PMC5624642
Lévesque N, Christensen KE, Van Der Kraak L, Best AF, Deng L, Caldwell D, MacFarlane AJ, Beauchemin N, Rozen R. Murine MTHFD1-synthetase deficiency, a model for the human MTHFD1 R653Q polymorphism, decreases growth of colorectal tumors. Mol Carcinog. 2017 Mar;56(3):1030-1040. doi: 10.1002/mc.22568. Epub 2016 Nov 1. PMID: 27597531.
Ben-Sahra I, Hoxhaj G, Ricoult SJH, Asara JM, Manning BD. mTORC1 induces purine synthesis through control of the mitochondrial tetrahydrofolate cycle. Science. 2016;351(6274):728–33.
Ju HQ, Lu YX, Chen DL, Zuo ZX, Liu ZX, Wu QN, et al. Modulation of redox homeostasis by inhibition of MTHFD2 in colorectal Cancer: mechanisms and therapeutic implications. J Natl Cancer Inst. 2019;111(6):584–96.
Liu F, Liu Y, He C, Tao L, He X, Song H, et al. Increased MTHFD2 expression is associated with poor prognosis in breast cancer. Tumour Biol. 2014;35(9):8685–90
Nilsson R, Jain M, Madhusudhan N, Sheppard NG, Strittmatter L, Kampf C, et al. Metabolic enzyme expression highlights a key role for MTHFD2 and the mitochondrial folate pathway in cancer. Nat Commun. 2014;5:3128
Cui X, Su H, Yang J, Wu X, Huo K, Jing X, Zhang S. Up-regulation of MTHFD2 is associated with clinicopathological characteristics and poor survival in ovarian cancer, possibly by regulating MOB1A signaling. J Ovarian Res. 2022 Feb 8;15(1):23. doi: 10.1186/s13048-022-00954-w. PMID: 35135596; PMCID: PMC8827288
EX4. Association with depression: The association between the MTHFR gene variant (rs1801133) and the occurrence of depression is an interesting finding. It is noteworthy that this association was observed in the control group but not in women with gynecologic cancers. Exploring the possible mechanisms underlying the association could provide further insights into the interplay between genetic factors and mental health outcomes.
Answer: Research showed that abnormal folate metabolism associated with the presence of mutated alleles of the 677C>T MTHFR polymorphism is associated with increased risk for depression [Nazki et al., 2014, Wu et al., 2013, Jiang et al., 2016]. Folate is essential for the biosynthesis of neurotransmitters associated with mood, stress, motivation and cognitive performance, such as serotonin (5-HT), noradrenaline (NA) and dopamine (DA) [Zhou et al., 2020]. Also folate deficient affects, homocysteine levels are higher, and research suggested a significant positive association between homocysteine levels and depressive-symptom severity [Bender et al., 2017].
Nazki FH, Sameer AS, Ganaie BA Folate: metabolism, genes, polymorphisms and the associated diseases. Gene. 2014 Jan 1;533(1):11-20. doi: 10.1016/j.gene.2013.09.063.
Wu YL, Ding XX, Sun YH, Yang HY, Chen J, Zhao X, Jiang YH, Lv XL, Wu ZQ. Association between MTHFR C677T polymorphism and depression: An updated meta-analysis of 26 studies. Prog Neuropsychopharmacol Biol Psychiatry. 2013 Oct 1;46:78-85. doi: 10.1016/j.pnpbp.2013.06.015.
Jiang W, Xu J, Lu XJ, Sun Y. Association between MTHFR C677T polymorphism and depression: a meta-analysis in the Chinese population. Psychol Health Med. 2016 Sep;21(6):675-85. doi: 10.1080/13548506.2015.1120327.
Zhou Y, Cong Y, Liu H. Folic acid ameliorates depression-like behaviour in a rat model of chronic unpredictable mild stress. BMC Neurosci. 2020 Jan 15;21(1):1. doi: 10.1186/s12868-020-0551-3. PMID: 31941442; PMCID: PMC6961331.
Bender A, Hagan KE, Kingston N. The association of folate and depression: A meta-analysis. J Psychiatr Res. 2017 Dec;95:9-18. doi: 10.1016/j.jpsychires.2017.07.019. Epub 2017 Jul 22. PMID: 28759846
EX5 – SNP-SNP interaction
Answer: Mutations in genes often occur during cancer progression, but the impact of a single nucleotide polymorphism in genome is small on cancer predisposition. Inheritance of multifactorial diseases does not occur, but it is known that relatives may be at higher risk of developing the disease. Statistical analysis of interactions between different loci is expected to clarify the etiology of multifactorial diseases. The effect of one locus may be ineffective alone or masked by effects at another locus, but the combined result of several SNPs may be significant. Thus, interactions play an important role in the study of gene function and may improve understanding of their importance in biochemical pathways. Some papers also analyze SNP-SNP interactions in relation to the etiology of gynecological cancers. Interactions of CYP1 gene variants [Li 2016] and X-ray repair of cross-complement 1 (XRCC1), tumor protein p53 (TP53) and fibroblast growth factor receptor 3 (FGFR3) gene polymorphisms [Liu 2019] was significantly associated with the risk of cervical cancer in Chinese women. Also in Chinese women, interactions of genes related to the metabolism of one-carbon units were analyzed. The joint impact of MTRR rs162036 and MTR rs1805087, MTHFR rs1801131 and MTHFR rs1801133, folate and MTHFR rs1801133 may contribute to breast cancer risk [Luo 2016].
Li S, Li G, Kong F, Liu Z, Li N, Li Y, Guo X. The Association of CYP1A1 Gene with Cervical Cancer and Additional SNP-SNP Interaction in Chinese Women. J Clin Lab Anal. 2016 Nov;30(6):1220-1225. doi: 10.1002/jcla.22006.
Liu GC, Zhou YF, Su XC, Zhang J. Interaction between TP53 and XRCC1 increases susceptibility to cervical cancer development: a case control study. BMC Cancer. 2019 Jan 7;19(1):24. doi: 10.1186/s12885-018-5149-0.
Luo WP, Li B, Lin FY, Yan B, Du YF, Mo XF, Wang L, Zhang CX. Joint effects of folate intake and one-carbon-metabolizing genetic polymorphisms on breast cancer risk: a case-control study in China. Sci Rep. 2016 Jul 12;6:29555. doi: 10.1038/srep29555. PMID: 27404801; PMCID: PMC4941723.
EX6 –additional discussion on the potential clinical implications and future research directions based on your results would be beneficial.
Answer: Certainly, further research is needed to better unravel the role of the MTHFD1 1958 A>G variant in ovarian cancer risk, as this protein may be a potential therapeutic target for future ovarian cancer therapies.
New 21 references have been added.

Round 2
Reviewer 1 Report
I think it can be accepted in the present form.